# An Original Methodology for the Selection of Biomarkers of Tenderness in Five Different Muscles

**DOI:** 10.3390/foods8060206

**Published:** 2019-06-11

**Authors:** Marie-Pierre Ellies-Oury, Hadrien Lorenzo, Christophe Denoyelle, Jérôme Saracco, Brigitte Picard

**Affiliations:** 1Université Clermont Auvergne, INRA, VetAgro Sup, UMR Herbivores, F-63122 Saint-Genès-Champanelle, France; marie-pierre.ellies@inra.fr; 2Univiversité de Bordeaux, Inria BSO, Inserm U1219 Bordeaux Population Health Research Center, SISTM Team, 33800 Bordeaux, France; Hadrien.Lorenzo@u-bordeaux.fr; 3Service Qualite des Carcasses et des Viandes, Institut de l’Elevage, 69007 Lyon, France; Christophe.Denoyelle@idele.fr; 4Inria BSO, CQFD Team, CNRS UMR5251 Institut Mathématiques de Bordeaux, ENSC Bordeaux INP, F-33400 Talence, France; jerome.saracco@math.u-bordeaux1.fr

**Keywords:** predictive model, tenderness, meat, biomarker, calpain, h2afx

## Abstract

For several years, studies conducted for discovering tenderness biomarkers have proposed a list of 20 candidates. The aim of the present work was to develop an innovative methodology to select the most predictive among this list. The relative abundance of the proteins was evaluated on five muscles of 10 Holstein cows: *gluteobiceps*, *semimembranosus*, *semitendinosus*, *Triceps brachii* and *Vastus lateralis*. To select the most predictive biomarkers, a multi-block model was used: The Data-Driven Sparse Partial Least Square. *Semimembranosus* and *Vastus lateralis* muscles tenderness could be well predicted (*R*^2^ = 0.95 and 0.94 respectively) with a total of 7 out of the 5 times 20 biomarkers analyzed. An original result is that the predictive proteins were the same for these two muscles: µ-calpain, m-calpain, h2afx and Hsp40 measured in m. *gluteobiceps* and µ-calpain, m-calpain and Hsp70-8 measured in m. *Triceps brachii*. Thus, this method is well adapted to this set of data, making it possible to propose robust candidate biomarkers of tenderness that need to be validated on a larger population.

## 1. Introduction

As emphasized by many authors, tenderness is considered the most important qualitative characteristic of meat. Tenderness is also a highly variable characteristic, and this wide variability appears to be a significant reason for consumer dissatisfaction and reduction in beef consumption. Therefore, tenderness inconsistency is a priority issue for the meat industry [1].

Tenderness can be evaluated by direct methods, such as instrumental or sensorial assessments, or by indirect methods, using muscular characteristics as tenderness predictors. Sensory methods are expensive, difficult to organize and time consuming [2]. In addition, these methods are invasive, and cannot be performed early enough to allow carcasses to be adapted to markets according to their level of tenderness. Thus, there is a need for method that is available early post mortem to estimate the tenderness potential of each carcass, and among the possible methods, the abundance of certain proteins is of particular interest for predicting tenderness [3,4]. Previous works reviewed by Picard et al. [5] made it possible to identify candidate biomarkers of tenderness. They belong to numerous biological pathways: heat shock proteins, oxidative and glycolytic metabolism, oxidative stress, muscle structure and contraction and proteolysis.

Predicting the overall tenderness of the whole carcass with a reduced number of indicators could be of interest for the beef meat chain. Indeed, with such information, retailers would be able to guide meat samples either to the traditional butchery circuits (if they are of good quality) or to the boning circuit (for those with a poor level of quality), thereby meeting consumer expectations. Therefore, it is interesting to investigate whether biomarkers of one muscle could predict the tenderness of another muscle of the carcass, and thus to evaluate the possibility of predicting the tenderness of a whole carcass by sampling only a reduced number of muscles.

Thus, the aim of the present work was to identify, from among a pool of 20 biomarkers of interest measured in five different muscles, whether a combination of biomarkers could predict the tenderness of each of the five muscles. Therefore, a specific statistical methodology adapted to a low number of samples with a large number of observations [6,7] was tested.

To investigate this question, we looked at five muscles that have been described in the literature as being muscles with different levels of tenderness. These were classified, according the Warner-Bratzler measurements, as tender (3.2 < Warner-Bratzler Shear Force < 3.9 kg; *Triceps brachii* (TB) or intermediate (3.9 < Warner-Bratzler Shear Force < 4.6 kg; *gluteobiceps* (GB)*, Vastus lateralis* (VL), *semimembranosus* (SM), *semitendinosus* (ST)) according to [8]. Indeed, data in the literature evaluating the correlations between the sensory tenderness of each of these five muscles and the “carcass sensory tenderness value” showed significant correlation coefficients (*p* < 0.0001) going from 0.75 for the ST muscle, to 0.81 to the VL muscle [9]. According to the literature, meat tenderness variation among muscles of the same carcass might be explained by animal genetics, feeding, handling or slaughter process [10], but also by muscle characteristics. Indeed, muscle background toughness is mainly determined by its organization and amount of connective tissue. This property is also influenced by the level of intramuscular fat, which is known to be highly variable across the muscles [11]. Then, the toughening and tenderization phases occur during postmortem storage of meat, as a result of sarcomere shortening during rigor development. The degree of contraction at which a muscle enters the state of rigor mortis is highly variable among different muscles within the carcass [12]. Thus, it might be supposed that the five muscles studied represent a diversity of physicochemical and muscular characteristics, which could be considered to be representative of the whole carcass.

## 2. Material and Methods

### 2.1. Animals

This study was conducted on 10 Holstein cows slaughtered between 29 and 90 months of age (56.5 ± 18.3 months on average) in a commercial slaughterhouse. The animals were slaughtered at an average carcass weight of 319 kg (±22 kg) and their carcasses were classified as 3 for fat score (European grade with a scale going from 1 to 5), with a conformation score going from P= to O-.

The slaughter was made in compliance with current ethical guidelines for animal welfare.

### 2.2. Muscle Sampling

In this experiment, we selected five muscles, which were excised for each animal and sampled for further analysis. A standardization of the sampling procedures was adopted for each muscle with a clearly identified sample location. The location of each muscle on the carcass can be seen in Figure 1.

Samples for biochemical (biomarkers and myosin heavy chains proportions) analysis were collected 15 min after slaughter, cut into small fragments, frozen in liquid nitrogen and packaged at −80 °C before grinding. Samples intended for sensory analysis were collected 24 h post mortem, and then they were vacuum-packaged and chilled for 7 days at +4 °C for ageing.

### 2.3. Biomarker Analysis

Total protein extractions were performed according to Bouley et al. [13]. The protein concentration was determined by spectrophotometry with the Bradford assay [14]. The evaluation of the relative abundance of protein biomarkers was realized by dot blot using antibodies previously validated as described in Guillemin et al. [15] by western-blotting (Table 1; [16]).

For dot-blot analysis, 15 μg of protein samples were spotted on a nitrocellulose membrane using Minifold I Dot-Blot apparatus. Dot-Blot membrane was air-dried during 5 min, and saturated with a 10% milk blocking buffer at 37 °C for 20 min. Then a primary antibody specific to the protein studied was incubated at 37 °C for 90 min. The anti-mouse fluorochrome-conjugated LICOR-antibody IRDye 800CW (1 mg/mL) diluted at 1/20,000 in 1% milk blocking buffer was hybridized for 30 min at 37 °C. Membranes were scanned by the Odyssey scanner (LI-COR Biosciences, Lincoln, NE, USA) and analyzed with GenePix (Axon Laboratory, Union City, CA, USA).

### 2.4. Myosin Heavy Chain Isoforms Proportions

The different types of myosin heavy chain (MyHC) isoforms were determined on the basis of previously determined migration patterns [17] using sodium Dodecyl Sulfate polyacrylamide gel electrophoresis (SDS-PAGE). This protocol was adapted from that of Talmadge and Roy [18]. Myofibrillar proteins were extracted from 200 mg of muscles with a buffer containing 0.5 M NaCl, 20 mM NaPPi, 50 mM Tris, 1 mM EDTA and 1mM DTT according to the protocol described in [19]. Then 5 µg of proteins were loaded in each well on a Mini-Protein II Dual Slab Cell electrophoretic system (Bio-Rad, Hercules, CA, USA). The separating gel consisted of 35% (*w*/*v*) glycerol, 9% (*w*/*v*) acrylamide-N,N′-methylenebisacrylamide (Bis) (50:1), 230 mM Tris (pH 8.8), 115 mM glycine, and 0.4% *w*/*v* SDS and the stacking gel of 47% (*w*/*v*) glycerol, 6% (*w*/*v*) acrylamide-Bis (50:1), 110mM Tris (pH 6.8), 6 mM EDTA, and 0.4% (*w*/*v*) SDS. The electrophoresis was performed at a constant voltage of 70 V for 30 h at 4 °C. At the end of migration, the gels were stained with Coomassie Blue [19] (Figure 2), and relative amounts of the different MyHC isoforms were quantified using ImageQuant TL v2003 software (Amersham Biosciences Corp., Piscataway, NJ, USA). 

### 2.5. Tenderness Evaluation

After ageing, the 5 muscles were trimmed and cut into 1.5 cm thick homogeneous steaks, then vacuum-packaged a second time and frozen at −20 °C until the sensory analysis. Meat in the form of 15 mm steaks was thawed and then cooked for 1 min 45 s in an Infra grill Duo Sofraca set at a temperature of 300 °C. After cooking, the steaks were cut into 20 mm cubes that were served on a plastic plate at an internal temperature of 55 °C. 

Sensory assessment was conducted by a 12-member panel selected, trained and controlled according to the XP V09-503 standard. The products were evaluated according to the conventional profile method (QDA) based on NF ISO 13300 [20]. Samples were presented in a monadic mode to the panelists.

The panelists evaluated the cooked samples for tenderness. Each attribute was rated on a 100-point non-graduated scale with a score of zero, on an ascending scale of quality for each attribute, being equivalent to tough, and a score of 100 being equivalent to tender. The sessions were carried out in a sensory analysis room equipped with individual boxes, under artificial non-colored lighting (Institut de l’Elevage, Villers-Bocage, France). 

Warner-Bratzler shear force on cooked samples (internal temperature of 55 °C) was measured according to the method of Schakelford et al. [21] with an INSTRON 3343 material testing machine. For each sample (each muscle of each animal), 10 measurements of shear force were made at different thicknesses of meat cores between 0.8 and 1.2 cm (Institut de l’Elevage, Villers-Bocage, France). 

### 2.6. Statistical Analysis 

Firstly, basic statistics, such as variance analysis and mean multiple comparisons with one factor and Pearson correlations were carried out using R-software [22]. 

Then, a specific statistical methodology adapted to a low number of samples with a large number of observations was tested. This method is a purely geometrical dimension reduction approach, which is suitable for managing a limited sample size with numerous variables stored in blocks. This makes it possible to provide interpretable information from the available data. However, it is important to mention that one should be wary of generalizing the results too quickly. Indeed, the generalization of the results would require a validation on a larger sample. Nevertheless, the proposed method will make it possible to clearly highlight some links that exist between the biomarkers and the force/or the tenderness on the available dataset.

#### 2.6.1. Methodology for Biomarkers Selection 

Among the different methodologies that can deal with multi-block structured data sets, the Data-Driven Sparse Partial Least Square (ddsPLS) shows interest for supervised problems whether in the case of regression (i.e., when the response variables are numerical) or classification (i.e., when the response variables are categorical). The ddsPLS approach allows variable selection in the covariate and in the output parts using two parameters, which are: *R*, the number of components in the model, and $L_{0}$, the maximum number of variables to be selected in the final model in the covariate part. These two parameters are fixed by cross-validation according to general supervised learning problem optimization solutions. The block structure can be designed according to many known factors, which are the time or the type of variable chosen for example.

Please note that, even if there are no missing values in the data set analyzed in the present work, the ddsPLS approach was initially developed to deal with missing samples, such as entire rows of missing values in given blocks.

#### 2.6.2. Partial Least Square (PLS)

Let X∈ℝn×p and Y∈ℝn×p be, respectively, the covariate matrix and the response matrix, describing *n* individuals through *p* resp. *q* variables. The *PLS* problem has been introduced by [18]. Its objective is to maximize the norm of the projected variance-covariance matrix YTX∈ℝq×p along the principal direction defined through u,v∈ℝp×ℝq with YvTXu∈ℝ being the chosen projection of individuals that must be maximized, under the unity constraints uTu=vTv=1. Finally, the optimization problem can be written as u,v=argmaxuTu=vTv=1YvTXu.

This makes it possible to define u and v as the weights of the first component of the model for the X part and for the Y part respectively. The nipals algorithm [18] makes it possible to find u and v in the same time.

Those two weights make it possible to define t=Xu∈ℝn and s=Yv∈ℝn, with the respective X and Y parts being first component scores of the model. In other words, they represent the individual positions in the dimension found.

Once that first component is defined, other dimensions are successively obtained thanks to the same kind of optimization. The deflation step makes it possible to remove the information from the current dimension to the residual data set. This step ensures that each component is orthogonal to the others.

The ddsPLS algorithm does not use the deflation step, since it has raised many questions for years [19]. It performs directly the *R*-dimensional Singular Value Decomposition (*SVD*) of the soft-thresholded variance covariance matrix. This solution does not permit the creation of more components than the rank of the soft-thresholded variance covariance matrix, which is itself majorized by the rank of the variance covariance matrix. In the high-dimensional context, this represents the number of variables in the Y part for multivariate regression, or the number of classes minus one in the classification cases.

#### 2.6.3. Data-Driven Sparse Partial Square (ddsPLS)

In the multi-block framework, the covariate part is now described by *T* blocks (different covariates matrices) Xt∈ℝn×pt,t=1,…,T describing the same *n* individuals. The ddsPLS algorithm, for a couple (R,L0) chosen by the user, begins with the *T SVD* of each of the variance-covariance soft-thresholded matrices SλYTXtn−1+. Sλ:x→signxx−λ+, where *sign* denotes the sign of a real, |.| denotes the absolute value, and .+ the max between its argument and 0. The parameter λ > 0 is chosen such as the cumulative number of non-null columns in the matrices SλYTXtn−1,t∈1,…,T is exactly equal to L0 and ∀∂λ∈ℝ+* the cumulative number of non-null columns in the matrices Sλ−∂λYTXtn−1+,t∈1,…,T is strictly higher than L0+1. Let Ut∈ℝpt×R,t∈1,…,T, denote the respective *R*-dimensional weights obtained through the corresponding *R*-dimensional *SVD*.

The next step makes it possible to gather that information through the super-weights, which are symbolized by βt∈ℝR×R. The scaled super-weights Utβt∈ℝn×R show the effect of one variable of block *t* on each of the super-components.

Finally, a third step makes it possible to build a regression model based on *T* regression matrices Bt, such as
Y≈∑t=1TXtBt∈ℝn×q

Using the Moore-Penrose pseudo-inverse, which does not imply matrix inversion, is particularly tricky in high dimensions or when $p_{t}>n$ in the context of regression. Let us mention that a linear discriminant analysis model is built using the *super-components* to predict new individual classes in the context of classification.

#### 2.6.4. Model Selection

As previously mentioned, the ddsPLS approach requires two parameters to be determined by the user: $L_{0}$, the maximum number of covariates (in the X part described by the *T* blocks) to be included in the model, and *R*, the number of components (calculated in the *T* underlying SVD), which is the analogous of the number of components in Principal Component Analysis.

Those parameters are determined according to the minimization of a chosen error risk. For the regression framework considered in this paper, the Mean Square Error in Prediction (*MSEP*) is naturally minimized over the response variables, i.e., the five tenderness variables. The error is computed on the Leave-One-Out (LOO) cross-validation predictions, denoted ${\hat{y}}_{m,i}$, for muscle $m\in \left\{VL,ST,TB,SM,GB\right\}$ and for carcass $i\in \left[1,10\right]$:$$\forall m\in \left\{VL,ST,TB,SM,GB\right\},MSEP\left(m\right)=\frac{1}{n}\overset{10}{\underset{i=1}{\sum }}{\left(y_{m,i}\;-{\hat{y}}_{m,i}\right)}^{2}$$

Since it has been chosen to standardize the Y matrix (zero-mean and unit-variance), when MSEP values are lower than 1, this reveals that the cross-validated models are better (on average) than just predicting the average of the selected sample at each iteration of the LOO cross-validation process, and thus the ddsPLS approach succeeds in retrieving links between the blocks of covariates and the corresponding response variables. On the other hand, when MSEP values are larger than 1, this indicates that predicting to the average of the selected sample at each iteration of the LOO cross-validation process provides more accurate predictions than the underlying cross-validated models, and thus the ddsPLS approach fails to retrieve information from the blocks of covariates and the corresponding response variables.

## 3. Results

### 3.1. Data Description

The five muscles were characterized by various slow oxidative MyHC I (*p* = 0.007) and fast oxido-glycolytic MyHC IIA (*p* < 0.001) proportions, whereas MyHC IIX (fast glycolytic) proportions were not significantly different among muscles (*p* = 0.129; Table 2).

The m. *semimembranosus* (SM) appeared to be the least slow oxidative muscle, with 10.5% of MyHC I, whereas the m. *gluteobiceps* (GB) and m. *Triceps Brachii* (TB), with proportions of MyHC I twice as large (20.4 and 22.5%, respectively), were the slowest oxidative muscles. The m. *semitendinosus* (ST) might be considered to be the most glycolytic muscle; even if the proportions of MyHC IIX were not significantly different among the five muscles, this muscle had low proportions of both MyHC I and MyHC IIA.

The TB muscle was the tenderest muscle, according the panelists. In accordance with this result, this muscle also had the lowest Warner-Bratzler shear force (Table 2). At the same time, the GB, which was the toughest muscle according to the mechanical analysis, had the lowest scores for sensory tenderness. These data are related to the significant correlation between tenderness scores and shear force values observed in this study (correlation: −0.53 when considering the five muscles together; *p* < 0.001). However, the correlation between tenderness and shear force is very muscle-dependent: for VL and ST muscles, the correlation is negative and significant (respectively −0.68 and 0.79), whereas for muscles GB, SM and TB, the correlation is non-significant.

Among the 20 analyzed biomarkers, three did not show a difference in relative abundances between the five muscles: αB-crystalin (*p* = 0.179), prdx6 (*p* = 0.293) and α-actin (*p* = 0.323) (Table 3).

The m. *Vastus lateralis* (VL) showed relative abundances significantly different from other muscles for many biomarkers. Indeed, it contained significantly less eno3, mylc1f, and Hsp40, and more m-calpain, pgm1, h2afx, and MyhcIIX than the other four muscles.

However, the VL muscle was close to the ST muscle for many biomarkers, especially Hsp70-1a, grp75 and mybph.

The SM muscle might be distinguished by a relatively high abundance of Hsp27, Hsp40, Hsp70-8, Dj1, sod1, mylc1f and µ-calpain.

The relative abundance of GB and TB muscle biomarkers was close, except for the Hsp20, which was lower in the TB muscle than in the GB muscle.

### 3.2. Links between Biomarkers and Tenderness Intra Muscles

When considering the tenderness of 1 muscle with associated biomarkers in the same muscle, Pearson correlations make it possible to highlight the following significant correlations (Figure 3):in the ST muscle, sensorial tenderness was negatively linked to Hsp40, Hsp27, pgm1in the SM muscle, sensorial tenderness was negatively linked only to mylc1fin the VL, GB and TB muscles, the sensorial tenderness was not significantly correlated with the abundance of any biomarkers (Figure 3).

When considering the correlations between shear force and protein biomarkers in the same muscle, Pearson correlations make it possible to highlight the following significant correlations:in the GB muscle, shear force was positively linked to prdx6in the TB muscle, shear force was negatively linked to prdx6 and mylc1f, and positively to Hsp27;in the other three muscles, the shear force was not significantly correlated with the abundance of an analyzed biomarker.

### 3.3. Links between Biomarkers and Tenderness among Muscles

When considering the correlation that appeared between the tenderness of one muscle and the biomarker abundance of the 4 other muscles, the highest number of significant correlations were observed for VL and SM tenderness with biomarkers measured in GB and TB muscles (Figure 4).

For biomarkers measured in GB muscle, the coefficient of correlation with VL tenderness were high particularly for h2afx (+0.85), m-calpain (+0.83), µ-calpain (+0.80), Hsp40 (+0.76), Hsp70-8 (+0.65) and for m-calpain (+0.77) and µ-calpain (+0.86), Hsp70-8 (+0.71) with SM tenderness.

For the biomarkers measured in TB muscle, high correlations were observed with VL tenderness for Hsp70-8 (+0.91), m-and µ- calpains (+0.77), as observed with biomarkers measured in GB muscle. For tenderness of SM muscle, high correlations were found with m- and µ-calpains (+0.89 and +0.71, respectively), Hsp70-8 (+0.63) as observed with VL tenderness, and with α-B crystalin (+0.63).

The tenderness of these two muscles VL and SM was especially correlated with biomarkers of GB and TB muscles.

Among these correlations, it might be noted that among biomarkers, µ-calpain, m-calpain, Hsp70-1a, Hsp70-8, tp53, pgm1 and mybph are always positively linked with tenderness, as opposed to MyhcIIx and Hsp40, which can be either positively or negatively linked to meat tenderness. Some biomarkers appeared to be more frequently correlated with tenderness: m-calpain, µ-calpain and Hsp70-8. These three biomarkers, evaluated in the TB and GB muscles, might be considered interesting biomarkers to predict meat tenderness.

Some coefficients of correlations were observed between biomarkers from TB muscle and tenderness of ST, SM and VL muscles (positively). However, biomarkers from TB muscles are mostly correlated with ST and TB shear force (negatively). It should be noted that many correlations can be observed with the tenderness of ST: dj1 (the highest; +0.76), MyhcIIx (in accordance with previous results in this muscle; +0.64), pgm1 (+0.58) and a-actin (+0.59) (Figure 4).

Many correlations were noted between the proteins measured in SM muscle and GB tenderness, whereas for the other muscles, only a few correlations were significant. The highest number of correlations measured in ST muscle was observed for the tenderness of GB muscle (Figure 4).

With regard to shear force, the muscle that is the most correlated with biomarkers is the ST muscle, the shear force of this muscle being especially correlated with biomarkers from VL, TB and GB muscles (Figure 5). The biomarkers that are significantly correlated with shear force are quite different from those found for muscle tenderness even if shear force and tenderness are significantly correlated in ST and VL muscles.

The details of the correlations obtained between the either tenderness of shear force of each of the five muscles and the 20 biomarkers of each muscle have been annexed in Figure 5.

### 3.4. Selection of Biomarkers the Most Predictive of Tenderness

To predict tenderness, the ddsPLS multi-block approach was used for five blocks of covariates $X_{t\;},\;t=1,\ldots ,5$ (each one corresponding to the pool of 20 biomarkers for each of the 5 muscles) and for one block of response variables Y (corresponding to the five tenderness scores, one score for each muscle). Note that in the model, we entered a 6th block, $X_{6}$, consisting of performance variables (age/weight). The covariates of $X_{6}$ were never selected in tenderness prediction, indicating that the protocol is well balanced, and that the variability of the tenderness is explained neither by weight or by age, nor by the combination of those factors.

Among all the proposed models, we selected the one that minimizes the *MSEP* (see Figure 6). Thus, the model with $L_{0}=8$ and R=2 was selected.

For the three muscles GB, ST and TB, the model indicated that the best way to predict tenderness is to predict the average value of the corresponding tenderness (i.e., the predicted values are equal to a constant, since there is no link with biomarkers). Hence, for these three muscles, we observe a horizontal alignment of the predicted values of tenderness, as shown in Figure 7.

For VL and SM muscles, the seven biomarkers that were selected by the model to predict global tenderness were derived from two different muscles: GB and TB. However, none of the selected biomarkers came from the muscle for which tenderness was predicted (Table 4). Among these seven biomarkers, two are derived from both GB muscle and TB muscle, leading to the number of distinct biomarkers solicited in the model being five.

The seven selected biomarkers measured in GB or TB muscles were the same for VL tenderness prediction and for SM tenderness prediction. These proteins belong to the heat shock protein family (Hsp40 and Hsp70-8) or were associated with the proteolysis (m-calpain and µ-calpain) and to the cellular response to DNA damage during oxidative stress (h2afx). Among the biomarkers selected in the model, two were selected in both the GB and the TB muscles: µ-calpain and m-calpain. The three other biomarkers are not the same; the model selected the h2afx and Hsp40 from the GB muscle and the Hsp70-8 from the TB muscle.

It is interesting to note that three biomarkers (see Figure 8) enter with a positive impact on SM and VL tenderness prediction: µ-calpain and m-calpain of the GB muscle and m-calpain of the TB muscle. Whereas the other biomarkers are associated with tenderness prediction either positively (in the VL muscle) or negatively (in the SM muscle), depending on the muscle in which it was measured: h2afx and Hsp40 of the GB muscle, Hsp70-8 of the TB muscle.

Thus, the contribution of biomarkers in the prediction of tenderness seems to be modulated according to the muscle.

The same analysis was performed by replacing the tenderness score with the shear force value. It appears that none of the shear force values were correctly predicted by the muscular biomarkers, regardless of the origin of the biomarkers. The model explained no more than 20% of the shear force, except for the TB muscle, where 75% of the shear force could be explained by 2 proteins: the eno3 coming from the GB muscle, and the mylc1f coming from the TB muscle. However, due to the poor quality of the model, it was difficult to draw conclusions on the relevance of these two biomarkers in a definitive way.

## 4. Discussion

To complete the approach presented here, several models were evaluated for various values of the parameters $L_{0}$ and *R*; the corresponding results are not provided here. One can observe that these models show good stability. More precisely, if we focused in the present paper on the case $L_{0}=8$ and *R* = 2, which makes it possible to minimize the mean of MSEP, we also studied cases where $L_{0}\in \left\{4,5,6,7,8,9\right\}$ (corresponding to the flat area of the MSEP curves in Figure 6). It appears that the biomarkers selected for $L_{0}=4$ are successively supplemented with 1, 2, 3 (and so on) biomarkers when selecting a maximum of 5, 6, 7 (and so on) biomarkers. Moreover, we also tried to explore the models based on a multi-block model with a block of response variables ***Y*** built only with the VL and SM tenderness, instead of the tenderness scores of the five muscles. The corresponding models selected the same biomarkers as previously indicated, and returned *R*^2^ values close to the values indicated in Figure 7. These different elements confirm the interest and relevance of the method, accurately predicting meat tenderness. The use of a multi-block model here (based on a linear relationship) allows us to highlight significant links between the tenderness and the abundance of certain biomarkers, even if they were not significantly correlated with tenderness. It is not inconsistent that the combination of several markers (individually little correlated with tenderness) can provide a relevant prediction of tenderness. It can also be noted that several biomarkers, significantly correlated with muscle tenderness, were not selected in the predictive models; this was certainly due to there being redundant information among different biomarkers. The two approaches, correlations and predictions, are thus fully complementary for resulting in relevant conclusions.

The highest number of significant correlations was observed for tenderness of VL and SM muscles with biomarkers measured in GB and TB muscles, mainly for VL tenderness with 12 proteins significantly correlated with tenderness, and 13 in SM muscle. Some coefficients of correlation were high both with SM and VL tenderness: for Hsp70-8, m-calpain and µ-calpain measured in GB and TB muscles. High correlations were found with VL tenderness for h2afx, and Hsp40 measured in GB and TB muscles. These proteins are the same that retained in the equations of prediction of the tenderness of these two muscles with the protein measured in GB muscle. Globally, the highest coefficients were found for VL and SM tenderness for most of the correlations: Hsp20, Hsp40, Hsp70-8, prdx6, dj1, MyBP-H, MyhcIIX, eno3, h2afx, pgm1, m- and µ-calpain. This suggests an important contribution of these proteins in the tenderness of these muscles.

Two models made it possible to predict the tenderness of a muscle (SM and VL) by the biomarkers of two other muscles of the carcasses (GB and TB). The selected proteins are the same in both predicted muscles, which testifies to the importance of these proteins in tenderness construction. Nevertheless, the coefficients associated with certain biomarkers highlight a different hierarchy in the selected biomarkers. Moreover, it might be noted that some coefficients might be reversed from one muscle to another. The fact that tenderness can be predicted only in the two muscles VL and SM can be explained by the fact that these two muscles are very different from the other three in terms of abundances of the analyzed proteins. The two muscles VL and SM are opposite for some proteins such as mylc1f and Hsp40, with the highest abundances being found in SM and the lowest in VL muscle. The abundances of m-calpain and µ-calpain are particular to these two muscles, in comparison to the three others, as m-calpain abundance was the highest in VL and µ-calpain abundance was the highest in SM muscle.

Both correlation and prediction analysis highlighted five proteins among the 20 putatively analyzed as interesting candidate biomarkers of tenderness: m-calpain, µ-calpain, Hsp40, Hsp70-8 and h2afx.

Among the selected biomarkers, the abundance of the m-calpain measured in GB and TB muscles enters the model with a positive coefficient in the prediction of the tenderness of the two muscles (VL and SM) and with a high effect. That highlights a major role of this protein in tenderness independent of the type of muscle. Meanwhile, the role of µ-calpain seems to be more muscle type-dependent.

Calpains (CAPN) are a large family of intracellular Ca^2+^-dependent cysteine-neutral proteases. The CAPN system is one of the endogenous proteolysis systems, and it plays a major role in meat tenderization [23]. Calpain’s actions are mainly due to having two major isoforms: μ-calpain and m-calpain. These two isoforms require different amounts of calcium in vitro, and their names suggest that they help promote the microscopic and milli-molar concentrations of intracellular Ca^2+^ [24]. Among CAPN enzymes responsible for meat tenderization, μ-calpain, which is encoded by the CAPN1 gene, plays a predominant role [25]. As far as we know, calpains can cleave limited myofibrillar proteins such as titin, desmin and vinculin, and contribute to the improvement of tenderness, whereas, high levels of calpastatin are related to decreased proteolysis and increased meat toughness [26,27]. Furthermore, both proteases were found to interact with several proteins belonging to different pathways: homeostasis, structure, glucose metabolism, heat stress, mitochondria, apoptosis. The different implication of calpains in meat tenderness observed in the present study could be explained by a different role of each calpain in tenderization. In accordance with this hypothesis, it was shown in lamb m. *Biceps femoris* that the autolysis of μ-calpain and loss of most of the activity occur within 7 days post-mortem whereas m-calpain activity did not decrease up to 56 days post-mortem. Moreover, very little post-mortem proteolysis is observed in the muscles of μ-calpain knockout mice suggesting its predominant effect on catalysis of proteins during postmortem aging.

Thus, we confirm that these proteases might be considered good biomarkers of beef tenderness, as already shown in the literature [16,28,29]. Moreover, we also confirmed, as previously shown by Guillemin et al. [16], that the contribution of proteolysis to tenderness might be different from one muscle to another. Indeed, these authors indicated that, in the LT muscle, m-calpain and µ-calpain are positively correlated with tenderness, whereas, in the ST muscle, tenderness appears to be positively correlated with µ-calpain, but negatively with m-calpain. These authors indicated that the inverse relation between tenderness and µ-calpain from one muscle to another is coherent with the inverse correlation between contractile and metabolic proteins and tenderness in ST and LT, and thus that muscular particularities might determine calpain contribution to tenderness. Thus, our results seem to confirm this hypothesis to the extent that the contribution of μ-calpain of TB in the construction of tenderness is positive in the VL and negative in the SM. Some studies analyzing a suppression of µ-calpain expression showed an increase in cell death through an activation of apoptotic caspases and an increase in Hsp27, Hsp70, and Hsp90 expressions. These data illustrate a link between the different biomarkers involved in tenderness prediction.

Heat Shock Proteins preserve cellular proteins against denaturation and possible loss of function [30]. A large set of Hsp have been associated with meat tenderness. For example, Guillemin et al. [28] proposed the ratio of Hsp70/small Hsps as a good predictor of muscle tenderness in charolais young bulls. The results of Bernard et al. [31], on the same cattle, found Hsp40 (DNAJA1 gene) to be a positive marker of beef toughness in ST muscle. Hsp40s represent a large protein family that functions as a co-chaperone protein of Hsp70. These two families, Hsp70 and Hsp40, are often co-localized in the same subcellular compartment. Hsp40s function as ATP-independent chaperones that bind non-native polypeptides and protect cells from stress by preventing protein aggregation. The major function of Hsp40 proteins is to regulate ATP-dependent polypeptide binding by Hsp70 protein. Among the Hsp70 family, the Hsp70-8 is known to slow down the process of cellular death and to protect tissues against oxidative stress.

According to Picard et al. [29], there is an inverse relationship between tenderness and proteins from the small Hsp family according to muscle type and breed. It is thus not astonishing that the influence of these proteins on tenderness might be reversed when considering two different muscles such as SM and VL. In accordance with these authors, the results of Guillemin et al. [32] showed different relationships between proteins of the interactome of tenderness and tenderness in two different muscles such as ST and LT. The results of the present study confirm these observations.

The last protein involved in the prediction of VL and SM tenderness in the present study was h2afx. H2afx belongs to the histone h2a family involved in the cellular response to DNA damage, notably during oxidative stress [33]. It has recently been shown that h2ax specifically controls the recruitment of DNA repair proteins to the sites of DNA damage [34]. In eukaryotic cells, one of the first cellular response is the phosphorylation of h2afx within 1 to 3 min after damage. The number of H2AX phosphorylated molecules, γ-h2ax, increases linearly with the severity of the damage and is necessary for the recruitment of other factors to the sites of DNA damage [35]. During stresses, h2afx recruits metabolism enzymes to activate energy generation, enhance protein synthesis, and recruit anti-stress proteins as hspa1a to protect cells from degradation. H2afx was proposed as a potential biomarker of beef tenderness by Guillemin et al. [16], who showed that three proteins, h2afx, sumo4 and tp53, were situated at the crossroads between groups of proteins involved in tenderness, as metabolism, structure, cellular stress, oxidative stress, apoptosis, and calcium signaling proteins. As h2afx modulates different pathways and ensure genome integrity [36], it could have a crucial role in meat tenderness, which is confirmed by the results of the present study. However, why its abundance only as measured in GB muscle is explicative of tenderness of SM and VL muscles, and why the relationships with tenderness are inverted in these two muscles, remain to be understood.

In conclusion, this study made it possible to propose an original statistical methodology well adapted to this type of multi-block data set with several covariates (biomarkers) larger than the number *n* of individuals (carcasses). Moreover, seven proteins could be proposed as candidate biomarkers of tenderness, but the understanding of the biological mechanisms involved according to the muscle considered, remain to be deeply analyzed.

## Figures and Tables

**Figure 1 foods-08-00206-f001:**
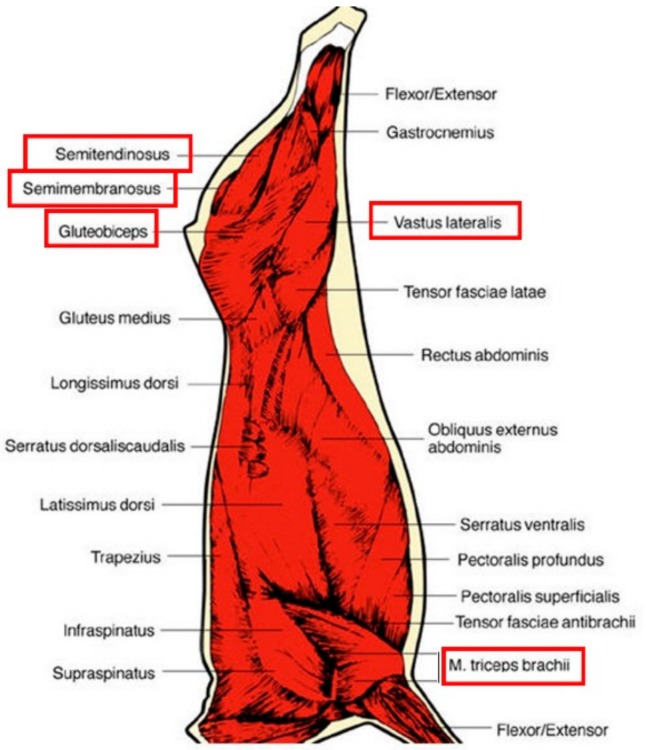
Location of the 5 muscles studied, the names of which are framed in red.

**Figure 2 foods-08-00206-f002:**
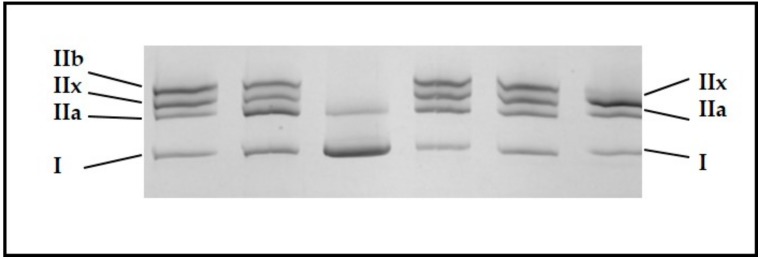
Illustration of the separation of the different myosin heavy chains isoforms depending on their molecular weight in different cattle skeletal muscles according to the protocol of Picard et al. [17].

**Figure 3 foods-08-00206-f003:**
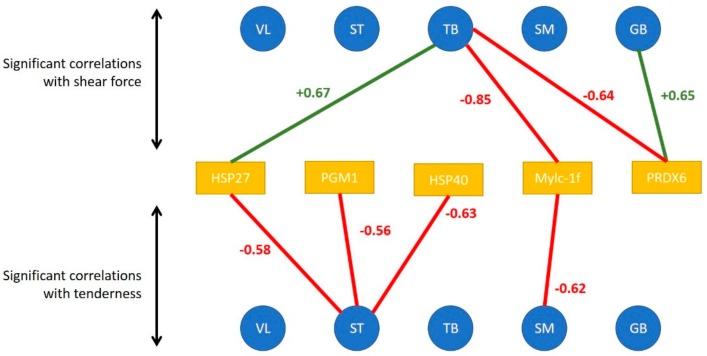
Significant correlations between tenderness/shear force of each muscle (represented by the blue disks) and the abundance of muscular biomarkers of the concerned muscle (represented by the orange rectangles). Red values correspond to negative correlations and green ones to negative correlations.

**Figure 4 foods-08-00206-f004:**
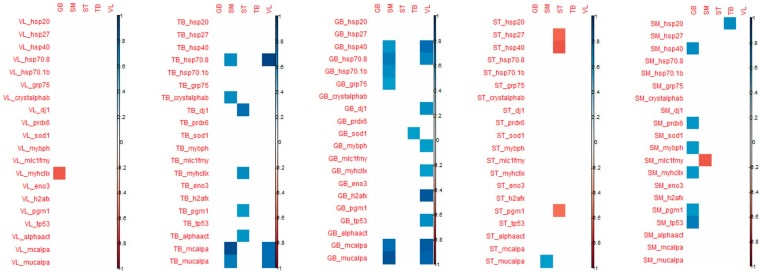
Correlations demonstrated between the tenderness of each of the 5 muscles (in column) and the 20 biomarkers of each muscle (in-line). Only the coefficients that were significant (corresponding to those that were superior or equal to 0.55/inferior or equal to −0.55) are indicated (positive correlations in blue and negative ones in red).

**Figure 5 foods-08-00206-f005:**
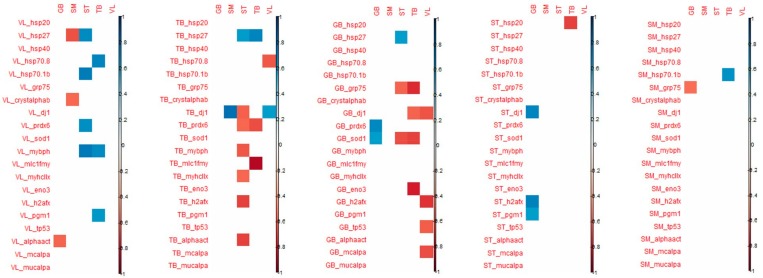
Correlations demonstrated between the shear force of each of the 5 muscles (in columns) and the 20 biomarkers of each muscle (in lines). Only the coefficients that were significant (corresponding to those that were superior or equal to 0.55/inferior or equal to −0.55) are indicated (positive correlations in blue and in negative ones red).

**Figure 6 foods-08-00206-f006:**
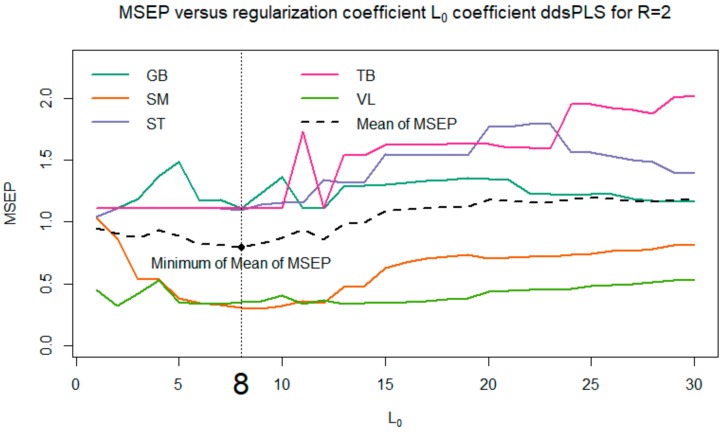
Evolution of the mean square error in prediction for each tenderness muscle and for the average tenderness depending on the maximum number of biomarkers selected in the model. In the present figure, the maximum is fixed at 30 biomarkers out of 100, i.e., 5 time 20 biomarkers.

**Figure 7 foods-08-00206-f007:**
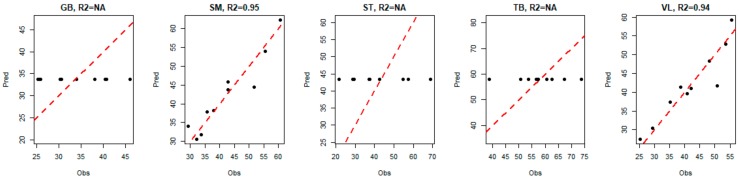
Evolution of predicted tenderness according to observed tenderness in each muscle.

**Figure 8 foods-08-00206-f008:**
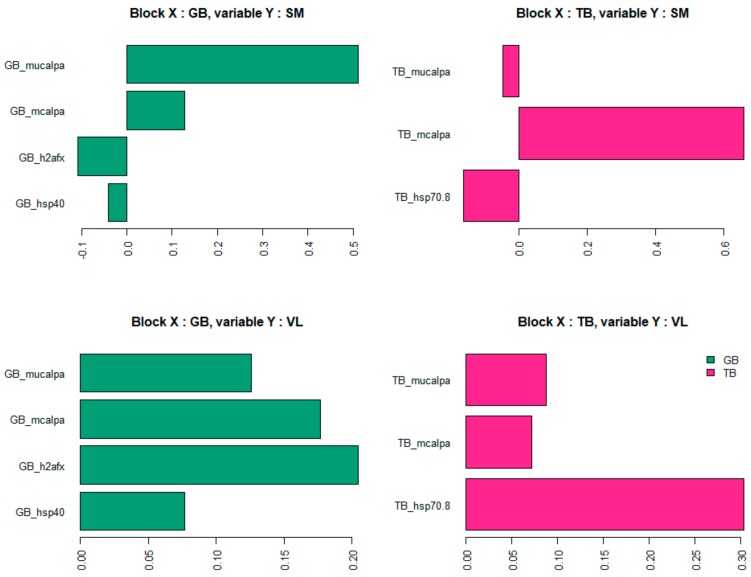
Coefficients associated with each biomarker in tenderness prediction models (representing the relative importance of each selected biomarker).

**Table 1 foods-08-00206-t001:** Gene names and protein names.

Biological Functions	Protein Names	Gene Names
Heat Shock proteins	αB-crystalin	CRYAB
	Hsp20	HspB6
	Hsp27	HspB1
	Hsp40	DNAJA1
	Hsp70-8	HspA8
	Hsp70-1b	HspA1B
	GRP75	HAPS9
Oxidative resistance	Dj1	PARK7
	*Cis-Peroxiredoxin*	PRDX6
	Super-oxide dismutase Cu/Z	SOD1
Structure	Myosin binding protein H	MyBPH
	Myosin light chain 1F (Mylc-1f)	MYL1
	Myosin heavy chain IIx	MyH1
	α-actin	ACT1
Metabolism	β-Enolase	ENO3
	Phosphoglucomutase 1	PGM1
Apoptosis and signaling	Tumor protein p53	TP53
	H2A Histone Family Member X	H2AFX
Proteolysis	m-calpain	CAPN2
	µ-calpain	CAPN1

**Table 2 foods-08-00206-t002:** Muscular characteristics of the 5 muscles.

Variables	GB	SM	ST	TB	VL	SEM	*p*
MyHC I (%)	20.4 ^b^	10.5 ^a^	16.8 ^ab^	22.5 ^b^	15.2 ^ab^	1.15	*p* = 0.007
MyHC IIA (%)	36.4 ^bc^	42.9 ^c^	25.8 ^a^	28.8 ^ab^	38.2 ^bc^	1.50	*p* < 0.001
MyHC IIX (%)	43.2	46.6	57.4	48.7	46.6	1.80	*p* = 0.129
Tenderness (on 100)	33.7 ^a^	42.3 ^a^	43.3 ^a^	57.9 ^b^	41.9 ^a^	1.85	*p* < 0.001
Shear force (daN)	11.0 ^b^	6.4 ^a^	7.4 ^a^	5.2 ^a^	6.8 ^a^	0.39	*p* < 0.001

Different letters indicate a significant difference (*p* ≤ 0.05) between the muscles. GB, *gluteobiceps*; SM, *semimembranosus*; ST, *semitendinosus*; TB, *Triceps Brachii*; VL, *Vastus lateralis*; SEM, standard error of the mean.

**Table 3 foods-08-00206-t003:** Relative abundance of each biomarker in the five muscles.

Biomarkers	ST	SM	GB	TB	VL	SEM	*p*
Hsp20	95 ^a^	150 ^b^	152 ^b^	99 ^a^	68 ^a^	7.7	<0.001
Hsp27	101 ^ab^	129 ^c^	118 ^bc^	111 ^bc^	86 ^a^	3.5	<0.001
Hsp40	118 ^b^	162 ^c^	112 ^ab^	117 ^b^	99 ^a^	4.0	<0.001
Hsp70-8	105 ^a^	141 ^bc^	123 ^ab^	129 ^abc^	156 ^c^	4.9	=0.012
Hsp70-1b	85 ^a^	188 ^c^	155 ^bc^	146 ^b^	81 ^a^	8.5	<0.001
GRP75	119 ^a^	153 ^b^	147 ^b^	155 ^b^	122 ^a^	4.0	=0.002
αB-crystalin	147 ^a^	195 ^a^	211 ^a^	153 ^a^	156 ^a^	10.3	=0.179
Dj1	95 ^a^	125 ^c^	110 ^abc^	114 ^bc^	101 ^ab^	2.8	=0.004
PRDX6	97 ^a^	110 ^a^	110 ^a^	110 ^a^	115 ^a^	2.7	=0.293
SOD1	79 ^ab^	95 ^a^	68 ^b^	62 ^b^	58 ^b^	3.7	=0.008
MyBP-H	102 ^a^	128 ^b^	140 ^b^	144 ^b^	101 ^a^	3.8	<0.001
Mylc-1f	78 ^ab^	109 ^c^	74 ^a^	89 ^b^	47 ^d^	3.6	<0.001
MyHCIIx	27 ^a^	33 ^a^	28 ^a^	29 ^a^	45 ^b^	1.3	<0.001
ENO3	103 ^a^	134 ^b^	121 ^b^	133 ^b^	83 ^c^	3.5	<0.001
h2afx	137 ^a^	173 ^b^	127 ^a^	139 ^a^	226 ^c^	6.4	<0.001
PGM1	121 ^a^	165 ^b^	132 ^a^	132 ^a^	306 ^c^	10.8	<0.001
TP53	106 ^a^	142 ^b^	94 ^a^	104 ^a^	153 ^b^	4.2	<0.001
α-actin	82 ^a^	89 ^a^	92 ^a^	96 ^a^	99 ^a^	2.7	=0.323
m-calpain	106 ^a^	139 ^b^	120 ^ab^	126 ^ab^	214 ^c^	6.8	=0.003
µ-calpain	123 ^a^	175 ^b^	136 ^a^	118 ^a^	134 ^a^	5.3	<0.001

**Table 4 foods-08-00206-t004:** Biomarkers selected of tenderness prediction.

Muscle in Which Biomarkers Were Measured	Biomarkers Selected for the Prediction of Tenderness	
	SM	VL
GB	µ-calpain (+) m-calpain (+) h2afx (−) Hsp40 (−)	µ-calpain (+) m-calpain (+) h2afx (+) Hsp40 (+)
SM	-	-
ST	-	-
TB	µ-calpain (−) m-calpain (+) Hsp70-8 (−)	µ-calpain (+) m-calpain (+) Hsp70-8 (+)
VL	-	-
*R* ^2^	0.95	0.94

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
