# Peer review of "An Original Methodology for the Selection of Biomarkers of Tenderness in Five Different Muscles"

_foods, 2019, doi:10.3390/foods8060206_

Reviewer 1 Report

This paper models the expression of 20 proteins in 5 tissues in 10 animals. Complex models are fitted to the proteomic data and the proteomic and statistical approach is interesting. However, regarding the sample size of n=10, this is a very small sample size, and it should be clearly emphasised that this is at the limits of interpretability. The authors should provide evidence, with citations, that their approach has sufficient statistical power to infer over the parameter space.

However, my main concern regarding the paper relates to the phenotypic data and how the muscles were treated. It is stated in the methods section, that the entire of the relevant muscles were excised at 15 minutes post mortem.

This is appropriate for the proteomic data and also for the fibre typing work.  There is no problem to examine correlations of protein expression across muscles sampled in this way and indeed the fibre typing experiments, but the relevance of interpreting the data as relating to objective shear force or sensory analysis on data collected on hot boned muscle is doubtful. I would have significant concerns regarding this approach, of removing entire muscles from the skeletal restraint within 15 minutes post-mortem, and then ageing and drawing any conclusions on their tenderness score. If the muscle is removed at 15 minutes, and aged off the bone, the muscle will contract, shorten and become irredeemably tough which is purely a physical process that has no relationship to the ultimate ‘true’ tenderness of the muscle treated appropriately. Appropriate tenderness analysis by either shear force or sensory methods requires the muscle to be left on the skeleton until rigor is achieved, i.e. until at least 24 hours post-mortem to prevent cold shortening. The phenotypic data on shear force and sensory data is not comparable to sensory or shear force data collected in an appropriate fashion. Hence the authors should insert clear and distinct caveats in their interpretation of this dataset to ensure the interpretation does not exceed the evidence base of the data.

Furthermore, the authors cite the paper of Wheeler et al 1997 as the reference for their method. However, they did not carry out their analyses in line with this cited paper. See extract below " Carcasses were chilled at 0°C for 24 h, then boneless strip loins (longissimus lumborum) were removed, vacuum-packaged, and aged 7 to 14 d at 2°C” there is a very clear and important difference in the protocols in followed in Wheeler et al and in the present study. This needs to be made clear.

“Samples. Twenty-seven steers, either 1/2 Piedmontese or 1/2 Bos indicus (all 1/2 British) were slaughtered humanely and dressed using typical procedures. Carcasses were chilled at 0°C for 24 h, then boneless strip loins (longissimus lumborum) were removed, vacuum-packaged, and aged 7 to 14 d at 2°C. These samples were obtained from USDA-ARS, Roman L. Hruska U.S. Meat Animal Research Center ( MARC) and used in order to provide a large range in tenderness.”

Regarding the paper’s writing, overall, the introduction is a little short, and assumes a lot of background information, and I think it would benefit the paper to make more explicit these considerations in the introduction. See points below.

CAN you include some introduction to the factors contributing to variation in tenderness among muscles? Line 66: this sentence should form part of the introduction and not the methods. Also ‘interesting’ should be defined. Please define better (in the intro – or indeed the discussion) what the interest is in these muscles.I agree it’s important to consider relative tenderness and relative toughness within a given muscle, and this is different from the different muscles. You have come up with some different cut off points for different muscles which I agree is important but without explaining the basis. Can you please explain a little more the basis for these different cutoffs?

Line 46: Please elaborate a little more on why it would be of benefit to predict tenderness for the whole carcass and why it is such a challenge.

Line 54: What is the basis for categorising these levels as tender/ tough. Have you carried out a comparison with sensory data? Are you comparing with the literature? Please elaborate on this important detail.

Minor points

Figure 4 is too small and difficult to read due to faintness of type

Line 39 : Rephrase ‘precocious’ to ‘available early post mortem’

Line 41: ‘abundance’ not abundancy

Author Response

Reply to the Associate Editor and the Reviewers

We would like to express our gratitude to the reviewers for their careful reading of the original manuscript and for their comments and suggestions. We thank the associate editor for the positive appreciation of our work, for allowing us to revise our manuscript and for giving us the opportunity to address the referee' comments. A point-by-point reaction on the reviewer' comments is provided below (in red).

We prepared a revised manuscript that, in our opinion, thoroughly addresses the reviewers’ comments and suggestions.

Reply to Reviewer #1

This paper models the expression of 20 proteins in 5 tissues in 10 animals. Complex models are fitted to the proteomic data and the proteomic and statistical approach is interesting. However, regarding the sample size of n=10, this is a very small sample size, and it should be clearly emphasised that this is at the limits of interpretability. The authors should provide evidence, with citations, that their approach has sufficient statistical power to infer over the parameter space.

In the introduction, we have given two references in order to provide evidence that our approach has sufficient statistical power to reach the objective of the analysis (see line 57). Indeed, in the analysis developed by (Lorenzo et al., 2019), the same statistical analysis was used in very similar framework (small number of samples and high-dimensional blocks with numerous variables).

The dimension reduction methodology developed here is a purely geometrical approach (without probability assumptions) and thus does not involve implicit inference. This method is suitable for managing a limited sample size and for providing interpretable information from the available data. It allows to clearly highlight some links between the biomarkers and the force / or the tenderness on our available data. However, it is important to be cautious about the interpretations allowed by our conclusions. Indeed, the generalization of the results obtained in this work would require a validation on a larger sample. These relevant remarks on interpretability and generalization of the results were added in the statistical analysis part (see lines 153-156).

However, my main concern regarding the paper relates to the phenotypic data and how the muscles were treated. It is stated in the methods section, that the entire of the relevant muscles were excised at 15 minutes post mortem.

This is appropriate for the proteomic data and also for the fibre typing work.  There is no problem to examine correlations of protein expression across muscles sampled in this way and indeed the fibre typing experiments, but the relevance of interpreting the data as relating to objective shear force or sensory analysis on data collected on hot boned muscle is doubtful. I would have significant concerns regarding this approach, of removing entire muscles from the skeletal restraint within 15 minutes post-mortem, and then ageing and drawing any conclusions on their tenderness score. If the muscle is removed at 15 minutes, and aged off the bone, the muscle will contract, shorten and become irredeemably tough which is purely a physical process that has no relationship to the ultimate ‘true’ tenderness of the muscle treated appropriately. Appropriate tenderness analysis by either shear force or sensory methods requires the muscle to be left on the skeleton until rigor is achieved, i.e. until at least 24 hours post-mortem to prevent cold shortening. The phenotypic data on shear force and sensory data is not comparable to sensory or shear force data collected in an appropriate fashion. Hence the authors should insert clear and distinct caveats in their interpretation of this dataset to ensure the interpretation does not exceed the evidence base of the data.

The description made in the material and methods is not totally accurate. Samples for biochemical measurements were taken within one hour after slaughter (around 15 minutes), but samples for sensory and mechanical analyzes were taken 24 hours after slaughter. Muscles removed were vacuum packaged and allowed to mature for 7 days at 4 ° C. This protocol is usually applied in our published experiments. The objective is to predict the final tenderness by measurements done at slaughter. The text was modified in order to better explain the exact procedure.

Furthermore, the authors cite the paper of Wheeler et al 1997 as the reference for their method. However, they did not carry out their analyses in line with this cited paper. See extract below " Carcasses were chilled at 0°C for 24 h, then boneless strip loins (longissimus lumborum) were removed, vacuum-packaged, and aged 7 to 14 d at 2°C” there is a very clear and important difference in the protocols in followed in Wheeler et al and in the present study. This needs to be made clear.

“Samples. Twenty-seven steers, either 1/2 Piedmontese or 1/2 Bos indicus (all 1/2 British) were slaughtered humanely and dressed using typical procedures. Carcasses were chilled at 0°C for 24 h, then boneless strip loins (longissimus lumborum) were removed, vacuum-packaged, and aged 7 to 14 d at 2°C. These samples were obtained from USDA-ARS, Roman L. Hruska U.S. Meat Animal Research Center (MARC) and used in order to provide a large range in tenderness.”

The article cited was not the good one. Instead of Wheeler et al., 1997, we have put the wright article: (Shackelford et al., 2004). This paper was cited for the mechanical measurements, the same steps as described in the article were applied for the measurements of shear force (the Warner-Bratzler shear force was estimated by the maximal shear at break with a Warner-Bratzler blade).

Regarding the paper’s writing, overall, the introduction is a little short, and assumes a lot of background information, and I think it would benefit the paper to make more explicit these considerations in the introduction. See points below. CAN you include some introduction to the factors contributing to variation in tenderness among muscles?

We have added some elements about the factors contributing to tenderness variation among muscle.

According to the literature, meat tenderness variation among muscles of a same carcass might be explained by animal genetics, feeding, handling or slaughter process (Warner et al., 2010) but also by muscle characteristics. Indeed, muscle background toughness is mainly determined by its organization and amount of connective tissue. This property is also influenced by the level in intramuscular fat, that is known to be highly variable across the muscles (Veiseth-Kent et al., 2018). Then, the toughening and tenderization phases occur during postmortem storage of meat, as a result of sarcomere shortening during rigor development. The degree of contraction in which a muscle enters the state of rigor mortis is highly variable among different muscles within the carcass (Rhee et al., 2004). Thus, it might be supposed that the 5 muscles studied represent a diversity of physicochemical and muscular characteristics that could be considered as representative of the whole carcass.

Line 66: this sentence should from part of the introduction and not the methods

Done

To investigate this question, we looked at 5 muscles that were described, in the literature, as muscles with different levels of tenderness. There were classified, according the Warner-Bratzler measurements, as tender (3.2 < Warner-Bratzler Shear Force < 3.9 kg; Triceps brachii TB) or intermediate (3.9 < Warner-Bratzler Shear Force < 4.6 kg; gluteobiceps GB, Vastus lateralis VL, semimembranosus SM, semitendinosus ST) according to (Belew et al., 2003). Indeed, data of the literature evaluating the correlations between the sensory tenderness of each of these 5 muscles and the “carcass sensory tenderness value”, showed significant correlation coefficients (p<0.0001) going from 0.75 for the ST muscle, to 0.81 to the VL muscle (Crosley et al., 1995). »

Also ‘interesting’ should be defined.

With the modification of the text the world ‘interesting’ disappeared. The reason why we considered these muscles as interesting is that they have different contractile and metabolic properties and also different tenderness

Please define better (in the intro – or indeed the discussion) what the interest is in these muscles. I agree it’s important to consider relative tenderness and relative toughness within a given muscle, and this is different from the different muscles. You have come up with some different cut off points for different muscles which I agree is important but without explaining the basis. Can you please explain a little more the basis for these different cutoffs?

These different muscles were chosen because they are known to have different contractile and metabolic properties and also different tenderness. Our objective was to create a variability in the tenderness scores.

Line 46: Please elaborate a little more on why it would be of benefit to predict tenderness for the whole carcass and why it is such a challenge.

Previous works have indicated that in a carcass, all the muscles have not the same evolution. For example, (J. Roberts et al., 2017; J. C. Roberts et al., 2017) indicated that while most meat from cow graded carcasses becomes less tender, within these carcasses, some muscles did not become tougher. Thus, knowing the level of tenderness in each muscle could be interesting for the meat chain as processors could potentially use these information as a guide for utilizing cuts which retain high eating quality and separating those which may require tenderness intervention to reach consumer acceptability.

We modified the text in order to add these elements:

“Predicting overall tenderness of the whole carcass with a reduced number of indicators could be of interest for the beef meat chain. Indeed, with such an information, the retailers would be able to guide meat samples either to the traditional butchery circuits (if there are of good quality) or to the boning circuit (for those with a poor quality level) and thus to meet consumer expectations.  That for, it is interesting to investigate whether biomarkers of one muscle could predict the tenderness of another muscle of the carcass, and thus to evaluate the possibility of predicting the tenderness of a whole carcass by sampling only a reduced number of muscles. »

Line 54: What is the basis for categorizing these levels as tender/ tough. Have you carried out a comparison with sensory data? Are you comparing with the literature? Please elaborate on this important detail.

These different muscles were chosen because they are known to have different tenderness according to the literature. We have indicated this element in the text.

To investigate this question, we looked at 5 muscles that were described, in the literature, as muscles with different levels of tenderness, classified as tender (3.2 < Warner-Bratzler Shear Force < 3.9 kg ; Triceps brachii TB) or intermediate (3.9 < Warner-Bratzler Shear Force < 4.6 kg; gluteobiceps GB, Vastus lateralis VL, semimembranosus SM, semitendinosus ST) according to (Belew et al., 2003)

Figure 4 is too small and difficult to read due to faintness of type

This figure was modified in order to be read easily.

Line 39 : Rephrase ‘precocious’ to ‘available early post mortem’

Done

Line 41: ‘abundance’ not abundancy

Done

Reviewer 2 Report

Overall the manuscript is well written and organized. The statistical design and methods followed are standard as used in meat science. Results are nicely presented and well discussed. This may help to improve the quality of paper before publication.

Change abstract and make it simpler and more comprehensible. If possible, avoid using information about statistics and design here, statistical part (section 2.6) is detailed enough.

Line no. 37 to 40, the premise is that sensory methods are time consuming, difficult and invasive. Is it OK to have it there in the introduction since you have used sensory evaluation in your study to support your findings?

Line no. 87, change ‘is’ to ‘was’, also at other places such as line no. 88.

Line no. 72, 134, 136, 154, Thanks to this….. thanks to that, avoid this language and write some simple and formal language.

Check whole manuscript for minor errors such as Line 214, butthis

There is no representative SDS-gel electrophoretogram in the paper.

Author Response

Reply to the Associate Editor and the Reviewers

We would like to express our gratitude to the reviewers for their careful reading of the original manuscript and for their comments and suggestions. We thank the associate editor for the positive appreciation of our work, for allowing us to revise our manuscript and for giving us the opportunity to address the referee' comments. A point-by-point reaction on the reviewer' comments is provided below (in red).

We prepared a revised manuscript that, in our opinion, thoroughly addresses the reviewers’ comments and suggestions.

Reply to Reviewer #2

Change abstract and make it simpler and more comprehensible: if possible, avoid using information about statistics and design here, statistical part (section 2.6) is detailed enough.

Done

Line no. 37 to 40, the premise is that sensory methods are time consuming, difficult and invasive. Is it OK to have it there in the introduction since you have used sensory evaluation in your study to support your findings?

What we want to say in the introduction is that sensory analyzes are expensive and difficult to apply routinely for predicting the quality of meat. Moreover, this measurement can be made only after aging and cooking, so cannot be applied for a diagnosis of sorting carcasses at the slaughterhouse. That's why we are looking for other ways to predict tenderness. However, in our study we used sensory analysis as a reference for having a value of tenderness to relate to the abundances of proteins measured to predict this score. At the end, the tool we would like to apply routinely to predict tenderness will be based on the measurements of proteins and not on sensory analysis.

Line no. 87, change ‘is’ to ‘was’, also at other places such as line no. 88.

Done

Line no. 72, 134, 136, 154, Thanks to this….. thanks to that, avoid this language and write some simple and formal language.

Done

Check whole manuscript for minor errors such as Line 214, butthis

Done

There is no representative SDS-gel electrophoretogram in the paper.

The figure has been added in the text.